# Effects of Hydrothermal Processing Duration on the Texture, Starch and Protein In Vitro Digestibility of Cowpeas, Chickpeas and Kidney Beans

**DOI:** 10.3390/foods10061415

**Published:** 2021-06-18

**Authors:** Prit Khrisanapant, Sze Ying Leong, Biniam Kebede, Indrawati Oey

**Affiliations:** 1Department of Food Science, Division of Sciences, University of Otago, P.O. Box 56, Dunedin 9054, New Zealand; prit.khrisanapant@outlook.com (P.K.); sze.leong@otago.ac.nz (S.Y.L.); biniam.kebede@otago.ac.nz (B.K.); 2Riddet Institute, Private Bag 11 222, Palmerston North 4442, New Zealand

**Keywords:** legumes, hydrothermal processing, texture, starch digestion, protein digestion, simulated digestion, kinetic modelling

## Abstract

Legumes are a vital candidate in the fight for food security as a sustainable and nutritious food source. The current study systematically investigated the effects of hydrothermal processing of varying durations (15–120 min) on the texture, starch and protein digestibility of cowpeas (*Vigna unguiculata*), chickpeas (*Cicer arietinum*) and kidney beans (*Phaseolus vulgaris*). Texture analysis and in vitro oral-gastro-intestinal digestion of each legume was combined with kinetic modelling to explore the rate and extent of their changes observed during hydrothermal processing. All three legumes showed rapid initial texture decay in the first 30 min of processing. Chickpeas showed the fastest rate of texture degradation with processing duration, whereas texture degradation of kidney bean was slower but reached the lowest hardness value among all beans when processed up to 120 min. The rate of starch and protein digestion increased with prolonged processing duration, whilst showing an inverse relationship with texture values. The extent of starch digestion continually increased with processing duration for all three legumes, whereas the extent of protein digestion decreased after 60 min in cowpeas. This study systematically demonstrated how choosing different processing times can modulate the rate of texture degradation, starch and protein digestion in legumes. The findings of this study can aid consumers and manufacturers on optimal processing to achieve the desired texture or modulate starch and protein digestibility.

## 1. Introduction

Legumes are a sustainable source of (plant-based) protein, carbohydrate, fibre and minerals. Legumes are consumed as a staple food in developing countries due to their economical cost and long (i.e., years) shelf life [1]. Legume consumption has been associated with lower risk of non-communicable diseases, such as diabetes [2]. Furthermore, legumes may be cultivated in poor soil and/or arid regions [3]; hence, legumes are a robust candidate to contribute to the world’s food security. Some examples of well-known legumes include cowpeas (*Vigna unguiculata*), chickpeas (*Cicer arietinum*) and kidney beans (*Phaseolus vulgaris*).

Despite the health and sustainability benefits, legumes are not as widely consumed in developed countries. This is due to a combination of factors, such as the involved process in their cooking, the potential of gastrointestinal discomfort from gas, undesirable odour [4] and generally lower protein digestibility (compared to animal sources) [5]. Complicating their utilisation is the presence of antinutrients which needs to be inactivated before consumption. Antinutrients include lectins, which can potentially cause food poisoning [6], α-galactosides which may be metabolised by gut bacteria to cause gastrointestinal discomfort [7], enzyme inhibitors [8], saponins [9], phytates [10] and tannins [11,12] which reduce starch and/or protein digestibility.

In an attempt to maximise the utilisation of legume seeds, hydrothermal processing (i.e., boiling) is the most widely adopted technique in domestic household cooking as well as in the food industry to be sold as canned products to consumers for convenience purpose. Hydrothermal processing softens a legume’s texture, so it is palatable for consumption. Moreover, reduction and/or inactivation of antinutrients has been known effective with hydrothermal processing [13]. Despite the importance of hydrothermal processing, the effects of processing on starch and protein digestibility have never been systematically studied in the literature, where a majority of studies employed only one or two different processing methods and/or intensity (time-temperature combinations) to process the legumes [14,15,16,17]. Furthermore, the measured starch and/or protein digestibility are usually expressed as a single in vitro or end-point value, with little or no information regarding the rate of digestion.

There is an emerging interest in utilising a kinetic approach in describing the starch and protein digestibility of foods [18,19,20,21]. This is because it allows for the approximation of the nutrient release and/or absorption within the gastrointestinal system as a function of time in response to pH changes and the presence of a wide of range of digestive enzymes. A slow, sustained release of starch and protein leads to prolonged satiety [22]. A slower rate of starch digestion also modulates blood glucose level, which is beneficial in reducing the risk of non-communicable diseases [23,24]. Conversely, an argument can be made with respect to a higher rate of protein digestion triggering a higher postprandial amino acid level, which may be desirable for athletes and the elderly [25,26]. Hence, the rate of starch and protein digestion could be a relevant information to promote legumes’ utilisation as part of daily diet.

The objective of the current study was to investigate the effects of hydrothermal processing duration (*t_c_* = 15–120 min) on three types of legumes, namely cowpeas (*Vigna unguiculata*), chickpeas (*Cicer arietinum*) and kidney beans (*Phaseolus vulgaris*). The three legumes were chosen based on three criteria: (i) each legume is widely cultivated in many regions of the world; (ii) each legume has a distinct macronutrient composition; and (iii) each legume was of different shape, colour and size. Hydrothermal processing was chosen because it is the most common and widely available processing technique to consumers regardless of socio-economic status. In this study, legumes processed at different hydrothermal processing duration were digested in an in vitro oral-gastric-intestinal process and the hydrolysis of starch and protein was followed as a function of digestion time (up to 6 h). The softening of legume texture along the hydrothermal processing duration was also measured.

## 2. Materials and Methods

### 2.1. Raw Material Handling and Storage

A batch of 7 kg commercial dried cowpeas (*Vigna unguiculata*), chickpeas (*Cicer arietinum*) and kidney beans (*Phaseolus vulgaris*) were purchased from a local market in Dunedin (New Zealand) in August 2017. Deformed, discoloured and damaged seeds were discarded and excluded from the experiment. The seeds were then vacuum-packed in opaque aluminium bags and stored in the dark at 4 °C until processing and analysis. Starch, protein, lipid and moisture content of each legume (Appendix A) was determined according to the glucose oxidase/peroxidase method (Section 2.5), Kjeldahl method [27] (Kjeltec 1002, Tecator, Hilleroed, Denmark), using a Soxhlet extractor [28] (HT1043 Soxtec, Tecator, Hilleroed, Denmark) and the gravimetric method with oven drying (Qualtex, Andrew Thom, Sydney, Australia), respectively.

### 2.2. Hydrothermal Processing of Legumes

Legume seeds (500 g) were soaked in distilled water at a seed-to-water ratio of 1:5 (*w*/*v*) for 16 h at 20 °C in an incubator (IL-11-4C, Lab Companion, Billerica, MA, USA), after which the soaking water was discarded. The seed’s moisture content increased from 9.13% to 58.24% during soaking due to water uptake overnight. The soaked legume was aliquoted (~100 g) and transferred into six individual stainless-steel sieve cages (Appendix A) to facilitate independent sampling at six different hydrothermal processing time points (i.e., *t_c_* = 15, 30, 45, 60, 90 and 120 min). A portion (~100 g) of the soaked legumes was set aside to represent an unprocessed state (*t_c_* = 0 min). Two large stainless-steel hydrothermal processing pots (10 L) and induction hot plates (2000 W, Micasa, Auckland, New Zealand) were used in this experiment, where three cages were assigned to each of them (i.e., cages for 15, 45 and 90 min in one pot and cages for 30, 60 and 120 min in the second pot). The legumes were boiled (~95 °C) in excess distilled water (≥1:20 (*w*/*v*) seed-to-water ratio). At each hydrothermal processing time point, the specified cage was removed from the pot and immersed in ambient tap water (~20 °C) to stop the thermal treatment whilst minimising starch retrogradation. The three legumes were then divided into two lots. The first lot was subjected to texture analysis and in vitro digestion within the same day (see Section 2.3 and Section 2.4 ).

The temperature–time profile of the seed interior during hydrothermal processing (identical setup as described above) was monitored using a thermocouple (1.5 mm diameter, Type K, Pico Technology, St. Neots, UK) and recorded every 5 s using a data logger (Pico Technology, St.Neots, UK). The time taken for the interior of the seed to reach 90 °C was approximately 30 min (Appendix A).

### 2.3. Determination of the Textural Properties of Boiled and Uncooked Legumes

Texture analysis of unprocessed, soaked, and boiled legumes were conducted using a compression test, as per Zhang et al. [29], with modifications. Hydrothermally processed legumes obtained from Section 2.2 were equilibrated to room temperature (~25 °C) for at least 1 h before texture analysis. Five seeds (or three for kidney beans) were placed onto the TA.HDplus texture analyser platform (Stable Micro Systems, Godalming, UK). A 50 mm diameter cylinder probe (P/50, Stable Micro Systems, Godalming, UK) was used to compress the seeds at a test speed of 10 mm/s, to 90% of its strain, into deformity. In this study, the legume seeds were compressed to 90%, instead of 50% of its height as recommended by Zhang et al. [29], because the cotyledon halves were prone to slide off each other with the compression probe rather than being compressed. The compression analysis for each legume was conducted in five independent replicates, utilising five seeds for each replicate (or three for kidney beans due to their larger size, and thus, exceeding the surface area of the cylinder probe in contact with the beans). Exponent software (Ver 6.1.16.0, Stable Micro Systems, Godalming, UK) was used to control the texture analyser and record the strain force over time during the compression test.

Two textural parameters were determined from each of the obtained texture profiles. Hardness is defined as the maximum force achieved during the compression test, represented in Newtons (N) [30]. Cohesiveness is the area under the curve of compression force versus time, expressed as Newton seconds (N.s).

At the end of each compression test, the disintegrated legumes were removed from the platform and probe using a clean spatula. The disintegrated samples were expected to mimic the way boiled legumes undergo a size reduction in the oral phase before being swallowed, and hence, were found to be suitable to be utilised for the subsequent in vitro digestion (Section 2.4). The disintegrated samples (Appendix A) produced from the compression test allowed consistency in the degree of mechanical deformity for different legume samples (due to hydrothermal processing duration, with and without hydrothermal processing) during the subsequent in vitro digestion procedure.

### 2.4. Simulated Oral-Gastro-Intestinal In Vitro Digestion of Uncooked and Boiled Legumes

Mechanically disintegrated legumes obtained from the compression test (Section 2.3) were immediately utilised for the in vitro oral-gastro-intestinal digestion on the same day to minimise any starch retrogradation that may occur. The legumes used in the in vitro digestion were boiled for 15, 30, 45, 60, 90 and 120 min. For comparison, unprocessed and soaked legumes were also investigated.

The in vitro starch and protein digestibility kinetics of compressed legumes were conducted according to the harmonised INFOGEST static in vitro digestion procedure [31]. The conditions of the oral, gastric and small intestinal digestion phases were simulated in vitro. During all simulated phases of digestion, the experiment was carried out at 37 °C in an incubator (7100, Contherm, Hutt City, New Zealand) with gentle agitation (55 stokes/min) (SK-R1807-S, DLAB, City of Industry, CA, USA) to approximate physiological conditions inside the oral-gastro-intestinal tract. In each in vitro digestion phase, electrolyte solutions mimicking digestion fluids were added.

Samples of digesta were withdrawn from the digestion vessels at different intervals throughout the 6 h-long in vitro digestion procedure to determine the degree of starch and protein digestibility. In the gastric phase, digesta were withdrawn at 0, 60 and 120 min for starch digestibility analysis, with additional time points at 30 and 90 min for protein digestibility. In the intestinal phase, digesta were withdrawn at 0, 20, 60, 90, 120, 180 and 240 min for both starch and protein digestibility analysis. Digesta allocated to starch digestibility analysis (0.5 mL each) were immediately subjected to a heat shock treatment at 100 °C for 10 min to stop activity of digestive enzymes [18]. Digesta allocated for protein digestibility analysis (0.3 mL) were mixed with 0.3 mL 20% (*w*/*v*) trichloroacetic acid (Ajax, Thermo Fisher Scientific, Auckland, New Zealand) to stop protein hydrolysis [32]. Samples were stored at −21 °C until further analysis.

### 2.5. Total Starch Measurement of Undigested Legumes Using a Glucose Oxidase/Peroxidase (GOPOD) Assay

The total starch content in boiled and uncooked legumes was quantified using the Megazyme Total Starch Assay Kit (K-TSTA-100A, Megazyme, Bray, Ireland) [33]. The Megazyme total starch kit contained all required enzymes. Frozen legume samples (~30 g, described in Section 2.2) were freeze dried (VirTis, SP Scientific, Warminster, PA, USA) prior to total starch measurement. Freeze-dried samples were ground using a laboratory blender (Waring, Auckland, New Zealand) for 60 s, with a pause every 30 s, at room temperature (20 ± 2 °C). The resulting flour was sieved to pass through a 180 µm mesh (Laboratory test sieve, woven wire, Endecotts, London, UK). The flour’s total starch measurement followed the kit’s instructions. To prevent overestimation of the total starch, the quantification of free glucose in each legume sample followed the same procedure, except each instance of enzyme addition was replaced with 100 mM of sodium acetate buffer (pH 5.0) instead.

### 2.6. Quantification of Free Glucose in the Legume Digesta Using a GOPOD Assay to Determine the Starch Digestibility of Boiled Legume

Legume starch digestibility was approximated through measurement of the D-glucose present in the aqueous phase of the digesta when hydrolysed using the Megazyme D-glucose Assay Kit (K-GLUC, Megazyme, Bray, Ireland). Digesta were gently thawed overnight at 4 °C and then centrifuged at 1090× *g* for 10 min at 4 °C (GPR centrifuge, Beckman, IN, USA) and supernatant was collected. D-glucose within the supernatant was measured according to the kit’s instructions. The amount of D-glucose released was converted by a factor of 0.9 (=162/180) into starch equivalents [33]. This information was combined with total starch values for undigested legume (from Section 2.5), and starch digestibility was expressed as the percentage of digested starch, see Equation (1):(1)$$\%\;\mathrm{of}\;\mathrm{digested}\;\mathrm{starch}=\frac{\mathrm{Free}\;\mathrm{glucose}\;\mathrm{in}\;\mathrm{digest}\;\times \;0.9}{\mathrm{Total}\;\mathrm{starch}\;\mathrm{content}\;\mathrm{of}\;\mathrm{undigested}\;\mathrm{sample}}=\frac{\mathrm{Starch}\;\mathrm{equivalent}\;\mathrm{in}\;\mathrm{digest}\;(\mathrm{mg})}{\mathrm{Total}\;\mathrm{starch}\;\mathrm{content}\;\left(\mathrm{mg}\right)}\;\times \;100\%$$

### 2.7. Quantification of Free Alpha Amino Groups in the Legume Digesta Using an O-Phthalaldehyde Assay to Determine the Protein Digestibility of Boiled Legume

Legume protein digestibility was measured using an *o*-phthalaldehyde (OPA) spectrophotometric assay based on the work of Lassé et al. [34]. Digesta were gently thawed overnight at 4 °C and then centrifuged at 15,300× *g* for 20 min. For each digestion time studied, 26.7 µL of supernatant was withdrawn and placed into a 96-well microplate (Greiner Bio-One, Auckland, New Zealand). An OPA solution (200 µL) containing 6 mM *o*-phthalaldehyde (Sigma-Aldrich, Steinheim, Germany), 50 mM sodium tetraborate (May and Baker, Dagenham, UK), 1.0% (*w*/*v*) sodium dodecyl sulfate (Thermo Fisher Scientific, Auckland, New Zealand), 2.0% (*v*/*v*) methanol (Ajax Univar, Thermo Fisher Scientific, Auckland, New Zealand) and 0.2% (*v*/*v*) β-mercaptoethanol (Sigma-Aldrich, Steinheim, Germany), prepared and used within 2 h of the assay, was added to the microplate. This resulted in a digesta-to-OPA solution ratio of 2:15. The microplate was placed inside a plate reader (Synergy 2, BioTek, Winooski, VT, USA) and the reaction was allowed to proceed with agitation for 30 s, and then without agitation for 90 s. Finally, the absorbance of the coloured solution was read at 340 nm. Solutions of L-serine standards (0–600 mg/L; Sigma-Aldrich, Steinheim, Germany) were used to prepare a standard curve, from which the hydrolysis of proteins and peptides within the digesta were calculated. Protein digestibility was then expressed as mg L-serine equivalent per g of legume sample dry weight. The digesta obtained at each time point were analysed in four replicates.

### 2.8. Data Analysis

The change in texture of the legumes at varying hydrothermal processing times was described with a first-order fractional conversion model, see Equation (2). To allow a fair comparison of the changes in the texture parameters as a function of hydrothermal processing time between the three types of legumes, the actual value of the texture parameters’ cohesiveness and hardness was converted and expressed as the percentage of the texture parameter with respect to the soaked bean (i.e., at processing time *t_c_* = 0). The converted experimental data were fitted to the model using SAS (Ver 9.4, SAS Institute, Cary, NC, USA) via a nonlinear regression procedure using the Marquardt maxiter fitting method from SAS. Two kinetic parameters (*k* expressed as × 10^−2^/min and *T_f_* expressed as a percentage) were estimated:(2)$${T=T}_{f}+\left(T_{0}-{\;T}_{f}\right)\times \exp^{\left(-{k\;\times \;t}_{c}\right)}$$
where T denotes the texture properties (percentage with respect to soaked bean) of legumes at hydrothermal processing time *t_c_* (min); *T_f_* indicates the proportional (%) texture properties of legumes at prolonged hydrothermal processing time; *T*_0_ represents the initial texture properties of uncooked legumes (100%, to represented by the textural properties of pre-soaked legume, *t_c_* = 0 min in this case); and *k* indicates the rate constant (×10^−2^/min) of texture changes/degradation due to hydrothermal processing. The standard error of the estimated kinetic parameters from the model was generated as part of the output from SAS.

During the 4 h-long in vitro small intestinal digestion phase, the changes in the starch and protein digested for each legume sample, boiled at varying durations, were also described using a first-order fractional conversion kinetic model, see Equation (3), using SAS software. Two kinetic parameters (*k_x_* expressed as × 10^−3^/min and *C_x_* expressed as a percentage of digested starch or mg/g of L-serine equivalent) were estimated:(3)$${C=C}_{x}+\left(C_{0\;}{-\;C}_{x}\right){\;\times \;\exp}^{\left(-k_{x\;\;}\times {\;t}_{d}\right)}$$
where C denotes the amount of starch (Equation (1), Section 2.6) or protein (Section 2.7) of a boiled legume sample being digested in vitro at time *t_d_* (min); *C_x_* indicates the asymptotic amount of digested starch (replace subscript *x* as *s*) or protein (replace subscript *x* as *p*) at prolonged small intestinal digestion; *C*_0_ represents the amount of digested starch or protein at the start of small intestinal digestion (*t_d_* = 0 min); and *k_x_* indicates the rate constant (×10^−3^/min) of starch (replace subscript *x* as *s*) or protein (replace subscript *x* as *p*) digestion. The standard error of estimated kinetic parameters model was generated as part of the output from SAS.

The fitting quality of the experimental data obtained from texture analysis and starch and protein digestion to the first-order fractional conversion model, see Equations (2) and (3), was evaluated by calculating the corrected R^2^, see Equation (4), and a visual examination of the residual (e.g., random distribution of residuals) and parity plots:(4)$${\mathrm{Corrected}\;R}^{2}=1\;-\;\frac{\left[\left(m\;-\;1\right)\left(1\;-\;\frac{{\mathrm{SSQ}}_{\mathrm{model}}}{{\mathrm{SSQ}}_{\mathrm{total}}}\right)\right]}{m\;-\;j}$$
where *m* is the sum of observations; *SSQ_model_* is the sum of squares of the regression model; *SSQ_total_* is the sum of squares of total observations; and *j* is the number of parameters estimated by the model.

For certain cooked legume samples exhibiting a linear starch and/or protein digestogram rather than following a first-order fractional conversion behaviour, a linear regression was used instead to estimate *k_x_*:(5)$$C_{L}={\;(k}_{x}{\;\times \;t}_{d}{)+C}_{0\;}$$
where *C_L_* denotes the amount of starch or protein of a boiled legume sample being digested in vitro at time *t_d_* (min); *C*_0_ represents the amount of digested starch or protein at the start of small intestinal digestion (*t_d_* = 0 min); and *k_x_* indicates the rate constant (×10^−3^/min) of starch (replace subscript *x* as *s*) or protein (replace subscript *x* as *p*) digestion.

Changes in the texture parameters as a function of *t_c_* and changes in starch and protein digestibility during gastric digestion were compared by one-way ANOVA for significant difference (*p* = 0.05) (IBM SPSS Statistics Ver 25, IBM, New York, NY, USA), followed by Tukey’s post hoc test.

## 3. Results and Discussion

### 3.1. Texture Changes of Cowpea, Chickpea and Kidney Bean at Different Durations of Hydrothermal Processing

Examination and comparison of the visual texture profile can give a rapid insight into the texture of legumes. According to Figure 1a, the texture profile of soaked chickpeas and kidney beans was characterised by a rough and uneven texture curve (especially chickpea), peaking at ~1800 N and ~2400 N, respectively. The roughness of the curve corresponds to cotyledon fracture (and movement of the deformed fractions) during compression, the fracturability of which can be approximated as crunchiness (Stable Micro Systems, 2020). This is especially pronounced in chickpeas, as this type of legume does not have a thick seed coat to modulate deformation during compression (unlike kidney beans). In contrast, the texture curve of soaked cowpeas appeared smoother with very minor troughs, also peaking at ~1800 N. This may be due to the smaller and flatter shape of a cowpea legume (Appendix A). When undergoing deformation, a smaller cotyledon has less volume to collapse, and therefore, deforms in a very uniform manner [35]. Hence, in a soaked state, it appears that the size and shape of the legume cotyledon dominates the texture profile by influencing the fracture behaviour of cotyledon.

During hydrothermal processing, the texture profile of all three legumes experienced two clear trends, illustrated in Figure 1b. Firstly, the maximum force (N) exerted on the legumes decreased down to ~250 N when boiled for 2 h. Secondly, the texture profile became smoother as the processing time increased, with the disruption and troughs becoming shallower and less frequent, indicating more uniform deformation (lower fracturability). Hence, the texture of legumes can be said to become less crunchy and smoother as hydrothermal processing proceeded. This makes sense, as deformation in uncooked plant materials tend to derive from cell fracture due to strong intercellular force (increased crunchiness), whereas processed material tends to result in cell separation (decreased crunchiness) [18,36,37].

From the texture profile curve, the cohesiveness and hardness of each uncooked and cooked legume were calculated. In all three legumes, cohesiveness and hardness showed a sharp decrease during the first 15 min of hydrothermal processing, with a sustained decrease until 45 min of hydrothermal processing in cowpeas and chickpeas, and up to 90 min for kidney beans (Figure 2). Hydrothermal processing for longer than 60 min did not appear to degrade the texture of cowpeas and chickpeas further, whereas the texture of kidney beans still degraded up to 120 min (Figure 2). Texture changes in these three legume types during hydrothermal processing were in agreement with Pallares Pallares, Loosveldt, Karimi, Hendrickx and Grauwet [18], who have also reported a large initial degradation in *Phaseolus vulgaris* after hydrothermal processing (95 °C) in excess of demineralised water. A good fit was found when using a first-order fractional conversion kinetic model to describe the texture changes of legumes as a function of processing time (Figure 2). A first-order fractional conversion kinetic model takes in consideration the non-zero texture value upon prolonged hydrothermal treatment. Using the fractional conversion model, the texture degradation that occurred during hydrothermal processing has been described in a variety of vegetables, including green beans, white beans, carrots, cucumbers, potatoes, peas, mushrooms, broccoli, squash, water chestnuts, beets and zucchini [36]. Recently, first-order fractional conversion model had also been used to describe the texture degradation in Bambara groundnut (*Vigna subterranean* (L.) Verdc.) during hydrothermal processing [19].

The kinetic parameters (rate of texture degradation; *k* and texture of legumes at prolonged hydrothermal processing time; *T_f_*), estimated using the first-order fractional conversion kinetic model, see Equation (2), to describe the texture degradation for the three types of legumes during hydrothermal processing are presented in Table 1. The *k* for cohesiveness and hardness was the highest in chickpeas, followed by cowpeas and kidney beans. In other words, the degradation rate (*k*) of the texture of cowpeas and chickpeas in terms of cohesiveness and hardness was approximately 1.78–1.94 and 2.13–2.55-fold higher than that of kidney bean, respectively (Table 1). In other words, hydrothermal processing degrades the kidney bean’s texture slower compared to the other beans that were investigated. Since the rate-limiting step for texture degradation can be attributed to the weakening of the middle lamella through pectin and cell wall polysaccharide solubilisation [37], the reason for the kidney bean’s slower texture degradation (lower *k*) for cohesiveness and hardness may be due to species-specific differences in the middle lamella and cell composition. Another factor that may have contributed to kidney beans’ texture retention is its larger size and homogeneity, compared to cowpeas’ smaller size and chickpeas’ irregular seed surface (Appendix A).

In all three legumes, it is important to note that the rate of cohesiveness degradation (*k*) during hydrothermal processing was higher than the rate of hardness degradation (Table 1). In other words, the overall consistency (cohesiveness) was decreasing faster than the peak maximum force that the legumes can withstand (hardness) during hydrothermal processing. This may be due to a shift from cell breakage to cell separation within the legume cotyledon during texture analysis as *t_c_* increases. At low *t_c_*, a strong intercellular adhesion existed [38]; however, as hydrothermal processing proceeds, cohesiveness rapidly diminishes as cells separate predominantly more than that they rupture and as pectin in the middle lamella solubilises, resulting in decreased intercellular adhesion [37,39].

At the end of hydrothermal processing (*t_c_* = 120 min), the cohesiveness for cowpeas, chickpeas and kidney beans dropped to 12.29, 12.29 and 7.61% of the initial value, respectively (Table 1). Hardness for cowpeas, chickpeas and kidney beans also dropped to 14.47, 15.68 and 7.94%, respectively. These estimated *T_f_* values (percentage of the proportion of the texture retained at long processing times) were calculated from the initial texture as detailed in Section 2.8. This illustrates that different types of legumes respond differently to varying durations of hydrothermal processing, with kidney beans generally retaining their texture better during initial stages of cooking (*t_c_* ≤ 60 min) than cowpeas or chickpeas.

The findings from this study also suggest that legumes used in this experiment do not exhibit the characteristics of a defect known as hard-to-cook (HTC), a typical condition in which legume seeds can harden and take a long hydrothermal processing time (>2–5 h) to soften [19]. The implications for changes in the texture after varying hydrothermal processing durations when relating to protein and starch digestibility will be further discussed in Section 3.4.

### 3.2. Starch Digestibility of Cowpeas, Chickpeas and Kidney Beans as Affected by the Duration of Hydrothermal Processing

#### 3.2.1. Starch Digestibility of Legumes during In Vitro Gastric Digestion

The amount of digested starch (%) of uncooked, soaked and hydrothermal processed legume samples at varying durations, during the 2 h-long in vitro gastric digestion is presented in Table 2. It was found that uncooked legume had exceptionally low starch digestibility (<6%) throughout the in vitro gastric digestion stage (Table 2), with the lowest starch digestibility observed for chickpea (<2%) starch digestibility. For soaked beans, the percentage of digested starch never exceed 7% for all the legumes. Interestingly, soaked chickpeas had increased starch digestibility compared to uncooked chickpeas. Perhaps this is due to water imbibition (hydration and swelling) making the matrix more accessible to enzymes. Moreover, the increased starch digestion could only be observed for chickpeas because they have been reported to contain lower amounts of digestion-inhibiting antinutrients (compared to cowpeas and kidney beans), which would have become activated during water imbibition [13,40].

When the hydrothermally processed legume samples were exposed to pepsin to simulate the in vitro gastric digestion phase (*t_d_* = 0 min), it was found that the amount of digested starch in cowpeas, chickpeas and kidney beans increased from 5.98% up to 12.55%, 6.56% to 10.59% and 2.89% to 5.77%, respectively, with increasing hydrothermal processing time between 15 and 120 min (Table 2). This indicates that hydrothermal processing contributes to greater susceptibility of starch to α-amylase hydrolysis during the previous oral phase, possibly through an enhanced enzyme–substrate interaction. This is not surprising, since hydrothermal processing is known to trigger starch gelatinisation, leading to greater susceptibility of starch granules to α-amylase hydrolysis [41,42,43].

At the end of the 2 h in vitro gastric digestion (*t_d_* = 120 min), a slight increase in the amount of digested starch was observed in hydrothermally processed cowpeas (up to 8.5%) and chickpeas (up to 1.2%), whereas for kidney beans, changes were very small overall (less than 1%) compared to when *t_d_* = 0 min (start of gastric digestion). A considerable increase in the amount of digested starch during the gastric phase is not expected since the only active hydrolytic enzyme present in the digestion vessel was pepsin, as oral α-amylase was previously deactivated by dropping the pH of the digestion mixture to 3. Any increase in the amount of digested starch during the gastric phase in cowpeas and chickpeas may be due to pepsin weakening the encapsulation of the storage protein-encapsulated starch granules. This phenomenon has been illustrated in common beans, wherein hydrolysis of storage protein (by pepsin in gastric phase; by trypsin and chymotrypsin in small intestinal phase) was illustrated to enhance starch digestion in vitro [44].

With respect to kidney beans, it is important to note that only a small increase in the amount of hydrolysed starch was observed, regardless of whether the kidney beans were cooked or not (Table 2). The amount of hydrolysed starch ranged between 3.5% and 5.96% for cooked kidney beans at the end of gastric phase. As mentioned above, the increase in starch digestion during the gastric phase may be attributed to the digestion of storage protein encapsulating starch granules, but perhaps this is insufficient to trigger an observable increase in starch digestion in the case of cooked kidney beans compared to cowpeas and chickpeas. This may be due to pepsin’s enzyme specificity; as Rovalino-Córdova, Fogliano and Capuano [44] reported, pepsin is capable of hydrolysing only ~5% of kidney bean protein after 6 h of in vitro gastric digestion. Therefore, it is possible that the pepsin utilised in this investigation does not effectively hydrolyse the storage protein present in kidney beans, and thus, the starch granules are still encapsulated within the storage protein (i.e., negatively affecting starch hydrolysis).

#### 3.2.2. Kinetics of In Vitro Starch Digestibility of Legumes at the Small Intestinal Phase

In this study, the final amount of digested starch for Uncooked and Soaked samples in all legumes were low (~9.5% to 23.0%). This is because the starch gelatinisation associated with boiled legumes had not occurred for the U and S samples, thus limiting the susceptibility of the starch granules to enzymatic hydrolysis [42]. In addition, the water soaking step could have activated antinutrients in the S sample, such as enzyme inhibitors, which can irreversibly bind to amylase and protease [45], reducing the starch digestibility. Hence, the kinetic behaviour of starch digestion during the in vitro small intestinal phase for both U and S samples were excluded from the remaining discussion.

Figure 3a displays the in vitro starch digestion kinetics of boiled legumes in the small intestinal digestion. A common trend was observed across all three types of cooked legumes, in which the starch was rapidly hydrolysed in the first 60 min of in vitro small intestinal digestion (*t_d_*) before slowing down and reaching a plateau. This is expected as the amount of (gelatinised) starch available to be hydrolysed by amylase (present in pancreatin) becomes rate-limiting. For legumes that were processed at lower hydrothermal processing times, however, it was found that their starch digestograms tended to exhibit a slow linear increase in the amount of hydrolysed starch. To quantitatively describe the starch digestion behaviour of the three legumes, a first-order fractional conversion kinetic model was applied to estimate the rate (*k_s_*) and the extent (*C_s_*) of starch digestion, see Equation (3). In cases where the samples were observed to not follow a first-order fractional conversion behaviour (in conjunction with a large standard error of the estimated kinetic parameter), a simple linear regression was utilised instead, see Equation (5). In this study, the changes in the starch digestibility of all cooked cowpea samples as a function of *t_d_* fitted well with the fractional conversion kinetic model. However, simple linear regression model was applied on chickpeas samples with a *t_c_* of 90 min or less, and on kidney beans samples with a *t_c_* of 60 min or less. The estimated kinetic parameters *k_s_* and *C_s_* for all boiled legumes are summarised in Table 3.

Based on Table 3, the rate of starch hydrolysis (*k_s_*) increases as a function of *t_c_* (hydrothermal processing time) for all legumes, suggesting an enhanced enzyme–substrate interaction between α-amylase and starch present in legumes that have undergone a longer hydrothermal processing. A higher estimated *k_s_* value indicates that more starch is being hydrolysed by α-amylase per minute at intestinal phase, and in the case of legumes subjected to hydrothermal processing, the starch gelatinised and would have increased their susceptibility to hydrolysis by α-amylase [42]. Therefore, it is likely that the rate of starch digestion increases as a function of hydrothermal processing time due to the increasing degree of starch gelatinisation. For example, the most dramatic increase in the estimated *k_s_* value was observed in kidney beans, increasing from 0.43 × 10^−3^/min when cooked for 15 min up to 6.93 × 10^−3^/min when cooked for 90 min, presenting a 16-fold increase in the starch digestion rate (*k_s_*).

Another explanation for the increase in the rate of starch digestion is likely through hydrothermally induced cell wall permeabilisation [21], which can increase the random diffusion of α-amylase onto the surface of starch granules [46], thus increasing starch hydrolysis. An increase in the cell wall permeability was qualitatively illustrated by Pallares Pallares, Alvarez Miranda, Truong, Kyomugasho, Chigwedere, Hendrickx and Grauwet [21], showing an increased diffusion of fluorescently labelled pancreatic α-amylase into the interior of isolated common bean cells (*Phaseolus vulgaris*) as a function of *t_c_*. Expectedly, their estimated *k_s_* also increased at prolonged *t_c_*, increasing from 0.65 to 1.30 × 10^−3^/min during *t_c_* of 30 min and 120 min, respectively. This is logical, as legume cotyledons naturally have encapsulated starch (and protein), restricting contact of digestive enzymes to their substrates. The rigid cell structure, especially of the cell wall, is a physical barrier to the diffusion of digestive enzymes into the cell, limiting hydrolysis of starch granules in vitro [24] and in vivo [47,48]. Therefore, it can be concluded that hydrothermal processing positively affects the rate of legume starch digestion (*k_s_*), likely through process-induced cell wall permeability.

An interesting finding with respect to boiled cowpeas was observed, where COW45, COW60 and COW90 samples appeared to share similar proportions of digested starch (between 65.29% and 66.90%) at the end of the small intestinal digestion (*t_d_* = 120 min). These samples also have overlapping predicted *C_s_* values, falling within 67.94% to 69.95% (Table 3). This may be because these samples fall within the temperature range for starch gelatinisation in cowpeas occur typically, which is reported to be between 79.13 and 84.83 °C [49]. In this study, the internal temperature of the seed took approximately 36 min and 48 min to reach 79 °C and 84 °C, respectively (Appendix A). Despite having similar amount of digested starch for the boiled cowpea samples (i.e., COW45, COW60 and COW90), their *k_s_* increased linearly with the *t_c_* of the samples, from 10.5 to 16.1 × 10^−3^/min. This result signified that the hydrothermal processing duration of cowpeas majorly influenced the rate of starch hydrolysis (*k_s_*) and achieved a similar final amount of starch being hydrolysed at the end of small intestinal digestion. If *C_s_* had been able to be predicted for chickpeas and kidney beans, perhaps a similar behaviour might have been observed near their respective gelatinisation temperature.

In contrast to boiled cowpea samples, the starch digestibility of chickpeas and kidney beans could only be observed to follow a first-order fractional conversion behaviour when they were cooked longer, i.e., 120 min for chickpea and above 90 min for kidney beans. At lower *t_c_*, the starch digestogram could be best described using linear regression (Figure 3a). A linear reaction (zero order reaction) is characterised by the rate of reaction being independent of the substrate concentration. A linear reaction model is generally described for reactions where only a small amount of product is formed from a substrate. In such a model, the substrate should be under such saturated conditions that its concentration remains effectively constant throughout the model, and thereby, the reaction rate appears independent of the substrate concentration [50]. In this investigation of chickpeas’ and kidney beans’ digestion, it is likely that due to the slow starch digestion rate (*k_s_*) at lower *t_c_* (probably due to incomplete starch gelatinisation), the concentration of starch (substrate) remains essentially constant throughout the experimental time frame. Hence, a zero-order model is used to describe the behaviour of chickpea and kidney bean samples, subjected to a shorter *t_c_,* during in vitro small intestinal digestion. In agreement with the study of Rovalino-Córdova, Fogliano and Capuano [44], a linear trend was also observed when investigating intact kidney bean cotyledon cell’s starch digestibility as influenced by proteases exclusion/inclusion. It is possible that if the digestion time were to be extended in this study, a digestogram, represented by a first-order fractional conversion, would eventually be obtained for chickpea and kidney bean samples subjected to a shorter *t_c_* as the concentration of available starch becomes rate-limiting. In other words, the experimental time frame only allows observation of the initial phase (linear part) of the expected starch digestogram. Of course, the in vitro small intestinal digestion was not investigated beyond 4 h in this study because the transit time for digesta passing through the small intestinal ileum is limited and usually occur within 4 h in the human body [51].

As illustrated in Table 3, the extent of starch digestion (*C_s_*) in cowpeas increased from 59.19% to 75.61% from when hydrothermal processing time (*t_c_*) was increased from 15 to 120 min. In kidney beans, the estimated *C_s_* can be seen increased from 56.37% to 66.92% when increasing *t_c_* from 90 to 120 min. Lastly, even though *C_s_* was not predicted for shorter hydrothermal processing time, chickpeas had the highest *C_s_*, attained at 94.09%, after 2 h of hydrothermal processing. The result suggested that a greater extent of starch digestion (*C_s_*) could be associated with higher hydrothermal processing time. A higher *C_s_* indicates that proportionally more of the starch granules within the legume cell was able to be accessed by α-amylase, which could be attributed to two major factors. First, hydrothermally induced cell wall degradation could have contributed to the increased *C_s_* at the end of 4 h-long small intestinal in vitro digestion. Secondly, hydrothermal processing could promote conformational changes (from protein denaturation) in storage proteins within the cell’s cytoplasmic matrix, thus degrading the encapsulating barrier between α-amylase and starch granules. This would explain an increase in the estimated *C_s_* for legumes that had been boiled longer.

Bringing together the findings, hydrothermal processing not only increases the rate of starch digestion of the studied legumes, but also the extent of digestion. The main contributing effect is most likely starch gelatinisation and induced cell permeability. Furthermore, while the general trend is the same, each legume, though containing a similar starch content, exhibited different responses to hydrothermal processing. Therefore, it was demonstrated that there are different nutritional implications for different legumes hydrothermally processed for the same duration.

### 3.3. Protein Digestibility of Cowpea, Chickpea and Kidney Bean as Affected by the Duration of Hydrothermal Processing

#### 3.3.1. Protein Digestibility of Legumes during the In Vitro Gastric Digestion

During the gastric phase of the in vitro digestion, there was no apparent trend in protein digestibility as a function of hydrothermal processing (*t_c_*) nor gastric digestion time (data not shown). The amount of L-serine equivalent (mg) released in the whole digesta per g dry weight was found to be no more than 11.35 mg for all legume type regardless of the duration hydrothermal processing (*t_c_*) and whether they were unprocessed or soaked. This may be due to pepsin’s enzyme specificity; as Rovalino-Córdova et al. (2019) reported, pepsin is capable of hydrolysing only ~5% of kidney bean protein even after 6 h of in vitro gastric digestion. Therefore, it is possible that the pepsin utilised in this investigation does not effectively hydrolyse the storage protein present in kidney beans, and thus, the storage protein continued to encapsulate starch (negatively affecting starch hydrolysis). Perhaps the poor specificity of pepsin towards protein in kidney beans is also extended to cowpeas and chickpeas. As such, under the experimental timeframe of 2 h gastric (pepsin) digestion, no discernible protein digestion could be observed. Thus, this section will only discuss results from the small intestinal phase of the in vitro digestion.

#### 3.3.2. Kinetics of the In Vitro Protein Digestibility of Legumes at the Small Intestinal Phase

During the in vitro small intestinal digestion, the protein digestogram of boiled cowpea samples exhibited a fast release of digested protein during the first 60 min of the in vitro small intestinal digestion, followed by a gradual plateau (Figure 3b). Conversely, the protein digestibility of chickpeas and kidney beans appeared to follow a linear trend. Notably, a visually linear in vitro protein digestogram of kidney beans was also obtained by Rovalino-Córdova, Fogliano and Capuano [44], although that particular study did not conduct any regression nor kinetic modelling. Utilising a similar approach as the starch digestibility in this study (Section 3.2.2), a first-order fractional conversion kinetic model was applied to estimate the rate (*k_p_*) and extent (*C_p_*) of protein digestion, see Equation (3). Again, where the first-order fractional conversion model is unable to predict a kinetic parameter, a linear regression was utilised where appropriate. The estimated kinetic parameters *k_p_* and *C_p_* for all boiled legumes are summarised in Table 4.

The results showed that hydrothermal processing duration (*t_c_*) has a positive impact on the rate (*k_p_*) of protein digestion in mostly all legumes (Table 4). The greatest trend of increase was observed in cowpeas, increasing from 7.40 to 15.60 × 10^−3^/min in COW15 to COW120 samples. Kidney beans also exhibited a significant increase in *k_p_* from 4.60 to 9.96 × 10^−3^/min in KID15 to KID120 samples. In chickpeas, *k_p_* increased only slightly as hydrothermal processing time increased beyond 90 min, as seen in the CHI90 and CHI120 samples, although the difference between the boiled chickpea samples was not significant at *p* = 0.05. A higher *k_p_* implies that protein is being hydrolysed faster by pancreatic proteases as an effect of hydrothermal processing. This is not surprising, as thermal treatment induces protein denaturation (and subsequent increased susceptibility to proteolytic enzyme hydrolysis). Another contributing factor may be that the proteolytic enzymes present in pancreatin also experienced enhanced diffusion through the cell wall, thus promoting the enzyme–substrate interaction [52]. Regardless, the increased rate (*k_p_*) and extent (*C_p_*) of protein digestion is likely attributed by protein denaturation [14,40], antinutrient inactivation [5,8,13] and enhanced process-induced cell wall permeability [21,44].

As previously mentioned, the most significant increase in *k_p_* was observed in kidney beans, followed by cowpeas. Removal and/or inactivation of antinutrient is very likely to have contributed to the increased rate of protein digestion [40]. For example, kidney beans are known to contain significant levels of protease inhibitors [8] and phytohaemagglutinin [53]. The protein constituent of this type of lectin resists proteolysis over a range of physiological pH [54], but it is denatured and, therefore, inactivated through heat [53]. Protease inhibitors are also inactivated by heat [8,55]. Likewise, the removal of digestion-hindering antinutrients such as tannins [12,56] or phytate [57] may have also contributed to the observed protein digestion improvement in kidney beans.

For all three types of legumes, the rates of protein digestion (*k_p_*) were very comparable for the samples boiled between 15 to 60 min, ranging between 6.09 and 8.72 × 10^−3^/min. This indicates that hydrothermal processing, *t_c_* of up to 60 min, did not appeared to affect *k_p_* to a considerable extent. This apparent lack of effect may be related to the presence of antinutrients and their incomplete inactivation within a short hydrothermal processing time. Indeed, it was demonstrated that approximately 45 min was required for the internal temperature of the seeds to reach 95 °C (Appendix A). Antinutrients are known to inhibit protein digestion through the direct inhibition of protease activity (e.g., in the case of trypsin and other protease inhibitors), through sequestration of ion cofactors (e.g., by phytate) or through forming irreversible complexes with protein molecules (as is the case for tannins) [56]. Khattab and Arntfield [13] have reported only partial removal of phytic acid in Canadian and Egyptian cowpea boiled for 35 min from 0.458 and 0.548 to 0.354 and 0.429 g/100 g, respectively; tannin also decreased from 14.71 and 5.78 to 6.30 and 3.28 mg/g, respectively. Therefore, the short *t_c_* may have led to incomplete inactivation of these antinutrients, which in turn could have contributed to the limited protein digestibility observed in this study. Moreover, the natural tightly packed cytoplasmic storage protein in legumes has been reported to limit water uptake, leading to incomplete protein denaturation when hydrothermally processed within a short time [44].

In contrast to the larger increase in *k_p_* observed in cowpeas and kidney beans, a very small increase was observed in all boiled chickpea samples, with *k_p_* ranging between 6.60 and 8.97 × 10^−3^/min in all treatments. This suggests that hydrothermal processing affects the rate of protein digestion in chickpeas to a lesser extent, compared to cowpeas and kidney beans (under experimental conditions conducted in the present study). This difference may be due to an interspecies difference, or the difference in the type and amount of antinutrients in each legume. Chickpeas have been reported to contain lower levels of antinutrients, such as protease inhibitors, compared to other species of legume [13,40].

The estimated extent of protein digestion (*C_p_*) at long digestion time (*t_d_* = ∞) could only be estimated for cowpeas with the first-order fractional conversion model; hence, the discussion will focus on cowpeas. At the end of the 4 h-long small intestinal digestion, cowpeas showed an increasing trend from an initial 33.25 mg L-serine equivalents released/g observed in COW15, increased to 46.04 mg/g in COW60, before showing a decreasing trend to 39.99 mg/g and 35.99 mg/g as observed in COW90 and COW120, respectively. The initially increasing *C_p_* trend observed in COW15, COW45 and COW60 is likely due to a combination of hydrothermally induced protein denaturation and the previously discussed inactivation/removal of antinutrients. The work of Naidoo, Gerrano and Mellem [14] has observed a great improvement in the extent of protein digestibility in five cowpea cultivars after hydrothermal processing, which is attributed to the hydrothermal inactivation of antinutrients. The disruption and unfolding of tightly packed storage proteins in cowpeas could have exposed sites to a proteolytic enzyme attack [58,59], hence possibly resulting in a higher *C_p_* compared to other legume types.

The observed increase in the estimated *C_p_*, as seen in COW60 followed by subsequent decrease in the estimated *C_p_*, suggesting that cowpea protein could be becoming less available for digestion after prolonged processing of 60 min (even if *k_p_* increases). This decrease may be due to heat-induced modifications such as protein insolubilisation and the creation of protein cross-links, as the partially unfolded peptides resulting from hydrothermal denaturation can subsequently aggregate to form complexes, which are resistant to enzymatic proteolytic attack [60]. This result is in agreement with Carbonaro et al. [61], who observed a reduction in in vitro protein digestibility for fava beans decreasing from 83.07% to 79.57% after 2 h of soaking and autoclaving at 120 °C for 20 min; this reduction was observed along with protein insolubilisation. Therefore, it is likely that the enhanced accessibility to proteolytic sites of denatured cowpea protein and inactivation of antinutrients is compromised by protein aggregation after a *t_c_* of 60 min for cowpeas.

### 3.4. The Relationship Between Texture and Starch and Protein Digestibility of Boiled Cowpeas, Chickpeas and Kidney Beans

In this study, it was evidenced that hydrothermal processing can cause microstructural changes, leading to loss of legume cohesiveness and hardness, as shown in Figure 1. Clearly, the impact on texture changes and digestibility of starch and protein in legumes can be vastly different depending on the hydrothermal processing duration. Therefore, the effect of an apparent change in the textural parameters of starch and protein digestibility at the small intestinal phase (represented as estimated kinetic parameters *k_s_* and *k_p_*, respectively) of legumes boiled at different durations are discussed in this section.

One of the key finding was that a decrease in cohesiveness and hardness of legumes due to prolonged hydrothermal processing could pose a major impact on the estimated rate of starch digestion (*k_s_*) (Figure 4a,b). The relationship between the texture parameters and the estimated kinetic parameters of starch and protein appeared exponential. In each legume, a major decrease in textural parameters (~50% of initial texture) resulted in relatively minor increases in *k_s_*. However, when the texture parameters were degraded at their lowest percentage to that of initial value when *t_c_* = 0 min (~15% for cohesiveness and ~20% for hardness of initial texture), a small change of texture in the legumes due to hydrothermal processing could result in a considerable increase of estimated *k_s_*. For example, the estimated *k_s_* for cowpeas hydrothermally processed within 15 and 30 min fall within a similar magnitude between 6.2 and 8.8 × 10^−3^/min (i.e., based on overlapping 95% confidence interval; Table 3), even though the texture degradation for cooked cowpeas at 15 and 30 min was major (at least 20% texture degradation) (Figure 4a,b). However, hydrothermally processing of any legume for more than 45 min has led to a steady increase in estimated *k_s_*. This illustrates that textural changes due to hydrothermal processing appears to have an exponential relationship to the rate of starch digestion. This may be due to the increased surface area from the increased cell separation (as informed by decreased cohesiveness and hardness) owing to solubilisation of the pectin-rich middle lamella [37,62].

Likewise, an exponential relationship was observed between the estimated protein digestion rate (*k_p_*) and legume cohesiveness and hardness (Figure 4c,d) regardless of legume type. Clearly, texture degradation through hydrothermal processing has a major impact on the estimated *k_p_*, where boiled samples with minimal texture degradation (i.e., at cohesiveness ~50% and hardness ~60% of the initial texture; *t_c_* = 15 min) tend to exhibit a lower estimated *k_p_* (~4–7 × 10^−3^/min). For boiled samples with moderate texture degradation (i.e., at cohesiveness ~20% and hardness ~25% of initial texture; *t_c_* = 30–60 min), these samples were predicted with modest *k_p_* values (7–8 × 10^−3^/min). Lastly, boiled samples with severe texture degradation (i.e., at cohesiveness and hardness ~10% of initial texture; *t_c_* ≥ 90 min) exhibited a faster estimated rate of protein digestion (*k_p_* > 8 × 10^−3^/min). In legumes, an overall lower texture parameter as an effect of hydrothermal processing appears to positively impact the estimated *k_p_*. Again, this may be due to an increased surface area for enzyme action, due to increased cell separation.

The duration of hydrothermal processing required for cowpeas, chickpeas and kidney beans to achieve a softened texture, without severe texture degradation, is likely to be different for each legume. For example, cowpeas and chickpeas to be processed for 30 to 45 min, and kidney beans are required to be processed for 45 to 60 min for them to attain similar hardness within the range of 20% to 25% of their soaked seed. According to Gwala et al. [19], legumes falling within this range are considered palatable for consumption. While increasing the processing time positively impacted the estimated extent of starch digestion at the end of small intestinal digestion (*C_s_*; Table 3), as well as the estimated rate of starch (*k_s_*) and protein (*k_p_*) digestion (Figure 4), for all legumes, cooked legumes underwent severe texture degradation (Figure 2). Finally, reduction in the estimated *C_p_* at the end of small intestinal digestion in cowpea becomes unavoidable at prolonged processing duration (Table 4). As such, the texture (degradation) of legumes due to hydrothermal processing highly influenced the estimated rate of starch (*k_s_*) and protein (*k_p_*) digestion. The nutritional implication is to simply enjoy legumes as a source of starch and protein; however, it is not critical to boil them until their texture is severely degraded. Conversely, to benefit from a slow, sustained starch and protein digestion, the duration of hydrothermal processing can be tailored to result in an appropriate texture for different legume types.

## 4. Conclusions

The systematic approach of studying legumes processed for varying duration has never been studied before, and the use of a suitable kinetic approach to describe the texture degradation, starch and protein digestibility is presented for the first time on cowpeas and kidney beans. The kinetic approach demonstrated how each legume responded differently to processing (*t_c_*) and during small intestinal digestion (*t_d_*), where the texture degradation of kidney beans was found to be slower than in cowpeas and chickpeas during processing, whereas the estimated rate of starch (*k_s_*) and protein (*k_p_*) digestion increased with increasing *t_c_*, but at different rates depending on the legumes. The estimated extent of starch digestion (*C_s_*) at the end of the 4 h in vitro small intestinal phase also increased with samples subjected to increased *t_c_*. In cowpeas, the estimated extent of protein digestion (*C_p_*) at the end of 4 h in vitro small intestinal phase decreased for samples with prolonged processing time (*t_c_* > 60). Future studies could explore the phenomenon potentially occurring during hydrothermal processing of legumes (e.g., starch gelatinisation, protein denaturation, antinutrient inactivation) to gain a mechanistic understanding. Our finding has created valuable insight into how legumes’ texture, as well as starch and protein digestion, may be affected by inter-species differences. In addition, a link between texture and the rate of nutrient digestion was observed. Overall, the starch digestion rate increased, as legumes lose their hardness and cohesiveness when boiled. In the practical perspective, this information could aid household and commercial processes to process legumes to a certain texture level to achieve the desired digestion characteristics, be that to maximise or modulate starch and/or protein digestion.

## Figures and Tables

**Figure 1 foods-10-01415-f001:**
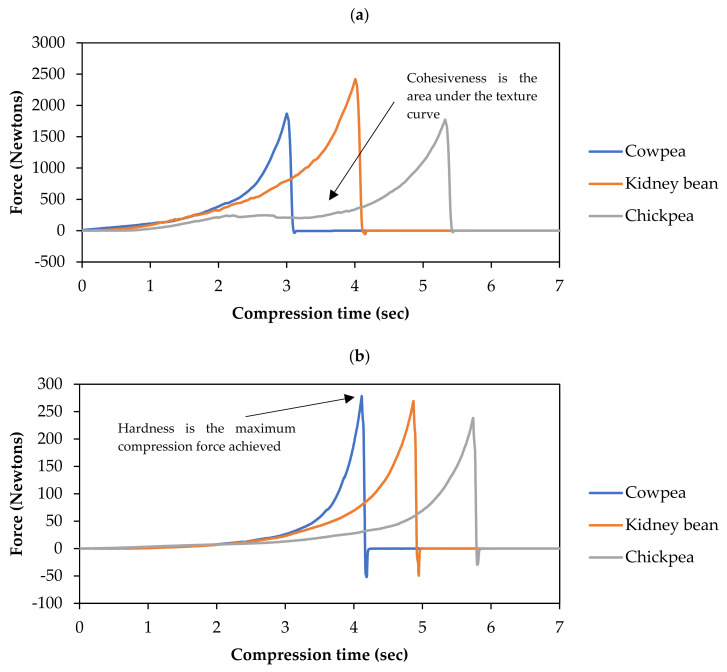
Representative texture profile curves of (**a**) soaked and (**b**) hydrothermally processed (*t_c_* = 120) cowpeas, kidney beans and chickpeas.

**Figure 2 foods-10-01415-f002:**
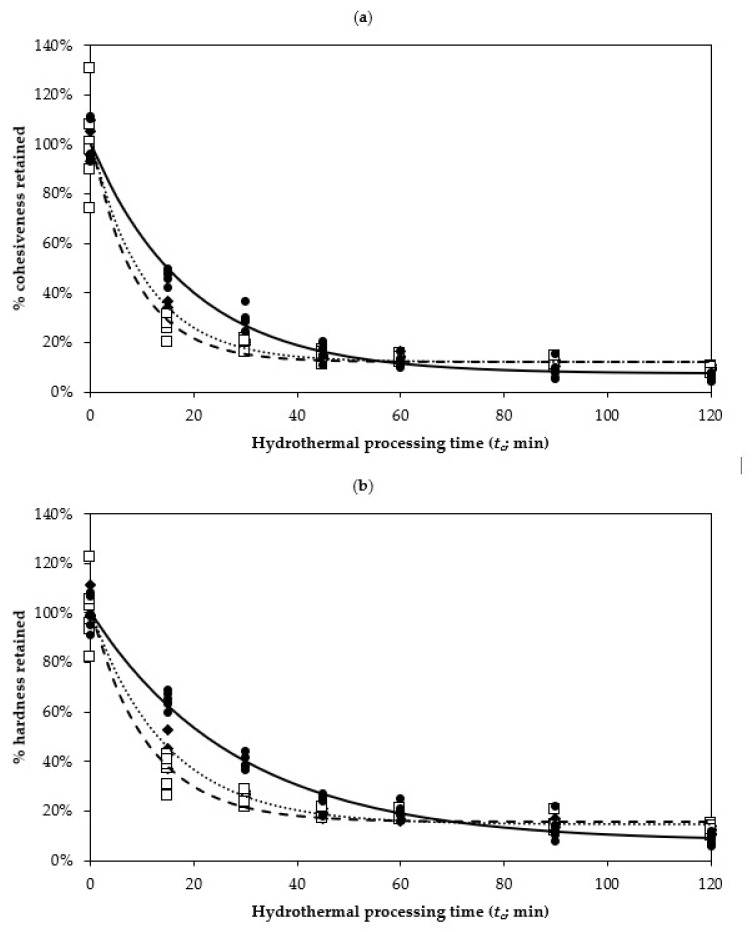
Changes in the (**a**) cohesiveness and (**b**) hardness of cowpea, chickpeas and kidney beans as a function of the hydrothermal processing time (min). Legend: ◆ 
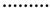
 represents cowpeas; 
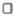


 represents chickpeas; 
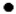


 represents kidney beans. The lines represent the fitted values by a first-order fractional conversion model, see Equation (2), to the hydrothermal processing time (N = 5 measurements).

**Figure 3 foods-10-01415-f003:**
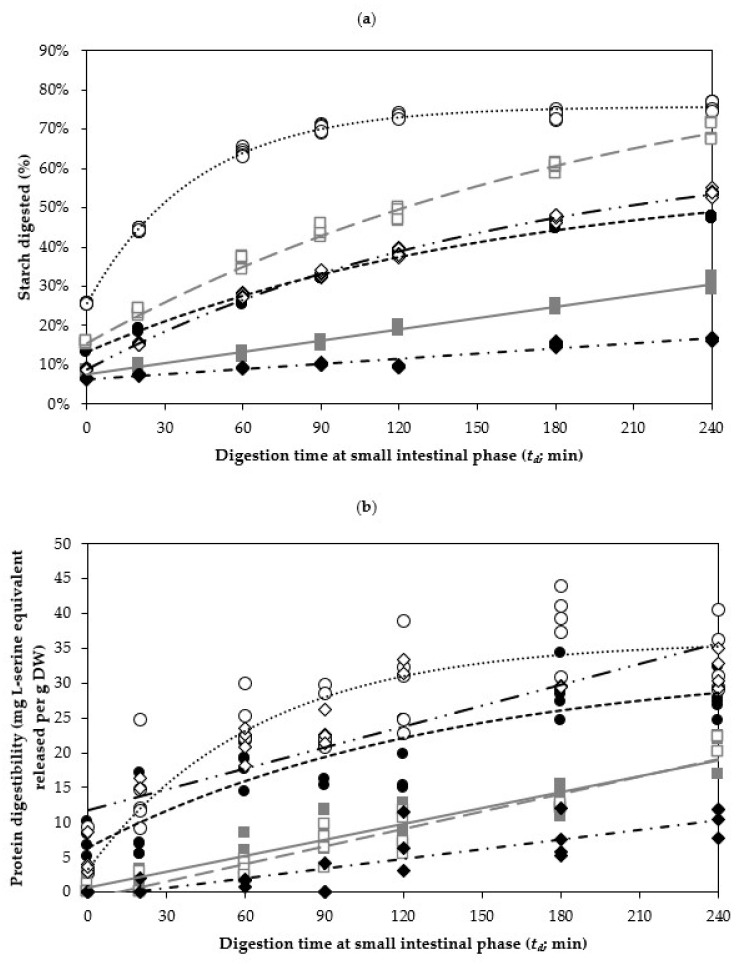
In vitro (**a**) starch and (**b**) protein digestion kinetic of boiled legumes in the small intestinal digestion. Markers indicate experimental values, while predicted values are shown as lines. The lines represent the fitted values by an appropriate kinetic model (Table 3 and Table 4). Legend: Cowpeas boiled for 15 min (COW15) 
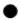

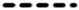
; Chickpeas boiled for 15 min (CHI15) 
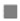

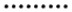
; Kidney beans boiled for 15 min (KID15) 
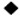

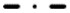
; Cowpeas boiled for 120 min (COW120) 
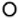

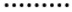
; Chickpeas boiled for 120 min (CHI120) 
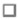

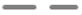
; Kidney beans boiled for 120 min (KID120) 
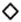


.

**Figure 4 foods-10-01415-f004:**
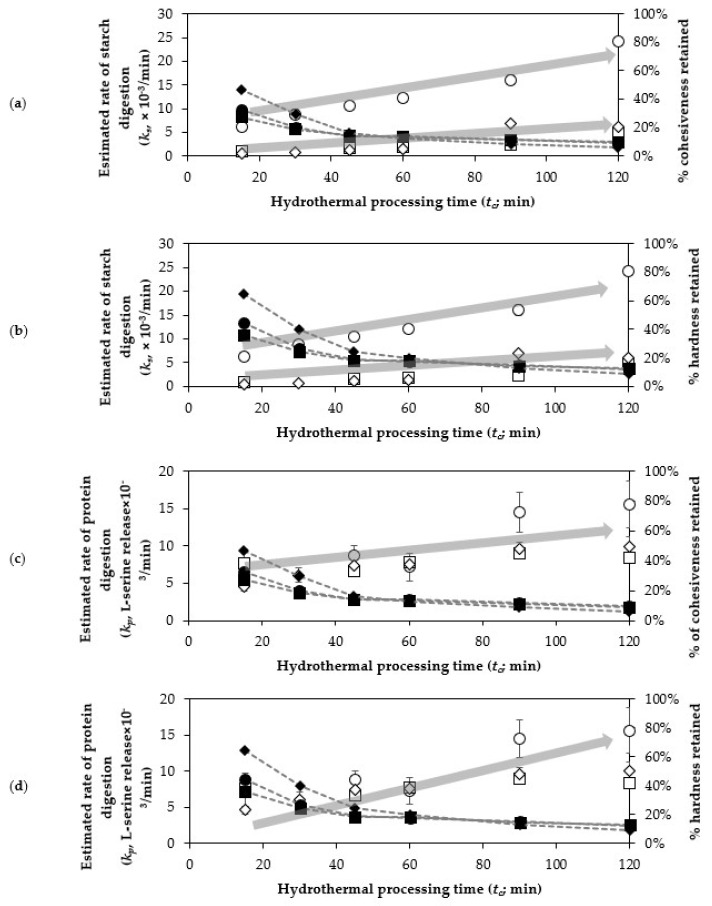
Relationship between the legume texture and the predicted digestion rate at the 4 h-long in vitro small intestinal phase of chickpeas, cowpeas and kidney beans as a function of hydrothermal processing time. From the top: (**a**) cohesiveness versus *k_s_* (**b**) hardness versus *k_s_* (**c**) cohesiveness versus *k_p_* (**d**) hardness versus *k_p_*. Circles represent the cowpeas; squares represent the chickpeas; and diamonds represent the kidney beans. Unfilled symbols represent the rates of digestion and filled symbols represent the texture values. Vertical error bars represent the approximate standard error of the estimated kinetic parameter.

**Table 1 foods-10-01415-t001:** Estimated kinetic parameters of legume cohesiveness and hardness as affected by varying durations of hydrothermal processing.

Texture Parameter	Legume	*k* (×10^−2^/min)	*T_f_* (%)	Corrected R^2^
Cohesiveness	Cowpeas	9.29 ± 0.47	12.29 ± 0.75	0.9941
	Chickpeas	11.16 ± 1.28	12.29 ± 1.41	0.9704
	Kidney beans	5.23 ± 0.25	7.61 ± 1.07	0.9906
Hardness	Cowpeas	6.78 ± 0.33	14.47 ± 0.90	0.9941
	Chickpeas	8.94 ± 0.77	15.68 ± 1.24	0.9812
	Kidney beans	3.50 ± 0.16	7.94 ± 1.32	0.9934

*k* represents the rate constant of the texture degradation during hydrothermal processing. *T_f_* represents the proportion (%) of the texture retained at prolonged hydrothermal processing times (with respect to initial texture at *t_c_* = 0 min). Values are expressed as the estimated kinetic value ± the standard error of the estimated kinetic parameters from the model (N = 5 measurements).

**Table 2 foods-10-01415-t002:** The amount of digested starch (%) in uncooked, soaked and boiled legumes during in vitro gastric digestion.

	Digestion Time (*t_d_*, min)	Hydrothermal Processing Time (*t_c_*, min)							
		Uncooked	Soaked	15	30	45	60	90	120
**Cowpea**	0	4.61 ± 0.26 ^A^ _a_	5.95 ± 0.15 ^A^ _b_	5.98 ± 0.96 ^A^ _b_	6.69 ± 0.12 ^A^ _c_	9.33 ± 0.18 ^A^ _d_	10.92 ± 0.11 ^A^ _e_	11.42 ± 0.27 ^A^ _f_	12.55 ± 0.10 ^A^ _g_
	60	5.62 ± 0.18 ^A^ _a_	5.90 ± 0.10 ^A^ _a_	9.57 ± 0.91 ^B^ _b_	9.48 ± 0.18 ^B^ _b_	14.67 ± 0.22 ^B^ _c_	20.49 ± 0.18 ^B^ _d_	18.80 ± 0.22 ^B^ _d_	19.03 ± 0.24 ^B^ _e_
	120	5.72 ± 0.22 ^B^ _a_	5.79 ± 0.06 ^A^ _a_	10.94 ± 0.30 ^C^ _b_	12.22 ± 0.20 ^C^ _c_	16.63 ± 0.22 ^C^ _d_	21.53 ± 0.29 ^C^ _e_	20.43 ± 0.33 ^C^ _f_	21.05 ± 0.37 ^B^ _g_
**Chickpea**	0	1.57 ± 0.10 ^A^ _a_	6.73 ± 0.24 ^B^ _b_	6.56 ± 0.21 ^A^ _b_	nd.	6.98 ± 0.26 ^A^ _b_	9.01 ± 0.28 ^A^ _c_	8.39 ± 0.24 ^A^ _d_	10.59 ± 0.27 ^A^ _e_
	60	1.95 ± 0.14 ^B^ _a_	6.58 ± 0.32 ^AB^ _b_	6.43 ± 0.23 ^A^ _b_	nd.	6.72 ± 0.32 ^A^ _b_	9.96 ± 0.19 ^B^ _d_	8.37 ± 0.22 ^A^ _c_	10.54 ± 0.38 ^A^ _e_
	120	1.54 ± 0.14 ^A^ _a_	6.24 ± 0.32 ^A^ _b_	7.67 ± 0.43 ^B^ _c_	nd.	7.82 ± 0.34 ^B^ _c_	10.35 ± 0.49 ^B^ _d_	8.90 ± 0.17 ^B^ _c_	11.82 ± 0.26 ^B^ _e_
**Kidney Bean**	0	3.61 ± 0.17 ^A^ _b_	3.61 ± 0.14 ^A^ _b_	2.89 ± 0.11 ^A^ _a_	4.19 ± 0.06 ^A^ _c_	5.13 ± 0.04 ^B^ _d_	5.91 ± 0.12 ^B^ _e_	6.45 ± 0.20 ^B^ _f_	5.77 ± 0.22 ^A^ _e_
	60	4.09 ± 0.18 ^B^ _a_	3.90 ± 0.1 ^B^ _a_	3.87 ± 0.13 ^B^ _a_	4.73 ± 0.07 ^B^ _b_	5.08 ± 0.05 ^B^ _c_	6.55 ± 0.12 ^C^ _f_	6.23 ± 0.20 ^B^ _e_	5.79 ± 0.09 ^A^ _d_
	120	3.80 ± 0.18 ^A^ _a_	4.83 ± 0.15 ^C^ _b_	5.96 ± 0.06 ^C^ _c_	5.04 ± 0.28 ^C^ _b_	3.50 ± 0.06 ^A^ _a_	5.09 ± 0.10 ^A^ _b_	5.70 ± 0.24 ^A^ _c_	4.95 ± 0.86 ^A^ _c_

Values are mean ± standard deviation (N = 6 measurements). Different uppercase letters (superscript) indicate a significant difference (*p* < 0.05) in the amount of digested starch as a function of digestion time (*t_d_*). Different lower-case letters (subscript) indicate a significant difference (*p* < 0.05) in the amount of digested starch as a function of hydrothermal processing time (*t_c_*). nd. not determined.

**Table 3 foods-10-01415-t003:** Estimated starch digestibility kinetic parameters for different boiled cowpeas, chickpeas and kidney beans during in vitro small intestinal digestion.

Legume	Sample	*t_c_* (min)	*k_s_* (×10^−3^/min)	*C_s_* (% Starch digested)	Corrected R^2^
Cowpea	COW15	15	6.20 ± 0.40 ^a^	59.19 ± 1.85 ^a^	0.9986 ^†^
	COW30	30	8.80 ± 1.00 ^ab^	61.49 ± 2.41 ^a^	0.9949 ^†^
	COW45	45	10.50 ± 0.30 ^b^	69.95 ± 0.61 ^b^	0.9996 ^†^
	COW60	60	12.20 ± 0.40 ^c^	67.94 ± 0.58 ^b^	0.9996 ^†^
	COW90	90	16.10 ± 1.10 ^d^	68.55 ± 0.97 ^b^	0.9980 ^†^
	COW120	120	24.10 ± 0.50 ^e^	75.61 ± 0.29 ^c^	0.9997 ^†^
Chickpea	CHI15	15	0.94 ± 0.02 ^a^	np.	0.9840 ^‡^
	CHI45	45	1.62 ± 0.03 ^b^	np.	0.9841 ^‡^
	CHI60	60	1.96 ± 0.03 ^c^	np.	0.9910 ^‡^
	CHI90	90	2.34 ± 0.05 ^d^	np.	0.9787 ^‡^
	CHI120	120	4.73 ± 0.34 ^e^	94.09 ± 3.68	0.9988 ^†^
Kidney bean	KID15	15	0.43 ± 0.02 ^a^	np.	0.9212 ^‡^
	KID30	30	0.74 ± 0.01 ^b^	np.	0.9952 ^‡^
	KID45	45	1.15 ± 0.02 ^c^	np.	0.9918 ^‡^
	KID60	60	1.44 ± 0.03 ^d^	np.	0.9797 ^‡^
	KID90	90	6.93 ± 0.45 ^e^	56.37 ± 1.70 ^a^	0.9980 ^†^
	KID120	120	6.07 ± 0.22 ^e^	66.92 ± 1.21 ^b^	0.9995 ^†^

*t_c_* represents hydrothermal processing time. *k_s_* represents the rate constant of starch digestion. *C_s_* represents the final amount of starch being hydrolysed at long digestion time. Values are presented as the estimates of the kinetic parameter ± the standard error of the estimated kinetic parameters from the model (N = 6 measurements). For each legume type, the estimated kinetic parameters with different superscripts in the same column indicate the difference based on their 95% confidence intervals. ^†^ Kinetic parameters were estimated using the first-order fractional conversion kinetic model, see Equation (3). Corrected R^2^ was calculated using Equation (4). ^‡^ Kinetic parameters were estimated using a linear regression, see Equation (5). R^2^ were generated as part of the linear regression. np. Not predicted, as a simple linear regression did not allow for the estimation of starch hydrolysed at a long digestion time.

**Table 4 foods-10-01415-t004:** Estimated protein digestibility kinetic parameters for different boiled cowpeas, chickpeas and kidney beans during in vitro small intestinal digestion.

Legume	Sample	*t_c_* (min)	*k_p_* (L-Serine Release × 10^−3^/min)	*C_p_* (mg L-Serine Equivalent/g DW)	Corrected R^2^
Cowpea	COW15	15	7.40 ± 2.28 ^a^	33.25 ^a^ ± 4.44 ^a^	0.9653 ^†^
	COW45	45	8.72 ± 1.34 ^a^	42.96 ^a^ ± 2.47 ^a^	0.9908 ^†^
	COW60	60	7.20 ± 1.84 ^a^	46.04 ^a^ ± 4.76 ^a^	0.9838 ^†^
	COW90	90	14.50 ± 2.64 ^a^	39.99 ^a^ ± 1.93 ^a^	0.9848 ^†^
	COW120	120	15.60 ± 3.16 ^a^	35.99 ^a^ ± 2.27 ^a^	0.9642 ^†^
Chickpea	CHI15	15	7.62 ± 0.55 ^a^	np.	0.8668 ^‡^
	CHI45	45	6.60 ± 0.67 ^a^	np.	0.7640 ^‡^
	CHI60	60	7.78 ± 0.71 ^a^	np.	0.8013 ^‡^
	CHI90	90	8.97 ± 0.67 ^a^	np.	0.8459 ^‡^
	CHI120	120	8.40 ± 0.45 ^a^	np.	0.9239 ^‡^
Kidney bean	KID15	15	4.60 ± 0.58 ^a^	np.	0.7290 ^‡^
	KID30	30	6.09 ± 0.94 ^ab^	np.	0.6378 ^‡^
	KID45	45	7.45 ± 1.14 ^ab^	np.	0.6308 ^‡^
	KID60	60	7.53 ± 0.79 ^b^	np.	0.7922 ^‡^
	KID90	90	9.58 ± 0.89 ^b^	np.	0.8302 ^‡^
	KID120	120	9.96 ± 1.36 ^b^	np.	0.7290 ^‡^

*t_c_* represents the hydrothermal processing time. *k_p_* represents the rate constant of protein digestion. *C_p_* represents the final amount of L-serine equivalent released after a long digestion time. Values are presented as the estimates of kinetic parameter ± the standard error of the estimated kinetic parameters from the model (N = 4 measurements). For each legume type, the estimated kinetic parameters with different superscripts in the same column indicate the difference based on their 95% confidence intervals. ^†^ Kinetic parameters were estimated using a first-order fractional conversion kinetic model, see Equation (3). Corrected R^2^ was calculated using Equation (4). ^‡^ Kinetic parameters were estimated using a linear regression, see Equation (5). R^2^ values were generated as part of the linear regression. np. Not predicted, as a simple linear regression did not allow for the estimation of starch hydrolysed at long digestion time.

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
