# Peer review of "Effects of Hydrothermal Processing Duration on the Texture, Starch and Protein In Vitro Digestibility of Cowpeas, Chickpeas and Kidney Beans"

_foods, 2021, doi:10.3390/foods10061415_

Round 1
Reviewer 1 Report
The manuscript presents a solid work done by the authors. The subject is interesting and important from a theological and nutritional point of view. The methods used are appropriate for the aim – especially the application of the standardized in vitro digestion model. However, a number of similar studies can be found, thus the authors should highlight the novelty of their work. Moreover, the manuscript appears to be extremely long, and some parts of the text can be deleted since they are some type of introductory paragraphs.
Detailed comments:
L81-84: Transferred to the conclusion section.
L95-100: The data can be added to supplementary materials.
Material and methods – this section should be shortened because it contains too much unnecessary introductory information. Please refer to the references. E.g. L274, L209 (if the authors follow the manufacturer recommendation and procedure, what is the reason to describe this in detail?), L167/181/189 (it is the standardized model – details can be found in the appropriate references).
Table 2: What is a potential reason that digested starch was not determined in chickpea at 30’?
L667-674: Example of the introductory paragraph that can be omitted (details are in the material and methods section).
Figure 3 – Figure Error?
Figures 3 and 4 can be combined.
Figures 5 and 6 can be combined with planes a, b, c, d.
L880-896 – Limitation and future directions – this section mainly summarizes the data and can be delated. Some future perspectives can be mentioned in conclusion.
Author Response
We thank Reviewer 1 for the positive feedback.
Point 1. L81-84: Transferred to the conclusion section.
Response: The novelty of the work has been transferred and integrated into the conclusion section. Please refer to Lines 796-799.
Point 2. L95-100: The data can be added to supplementary materials.
Response: The data has been removed from the main text into the supplementary material as Table S1.
Point 3. Material and methods – this section should be shortened because it contains too much unnecessary introductory information. Please refer to the references. E.g. L274, L209 (if the authors follow the manufacturer recommendation and procedure, what is the reason to describe this in detail?), L167/181/189 (it is the standardized model – details can be found in the appropriate references).
Response: Thank you for the comment. We have omitted the detailed information describing the “Simulated oral-gastro-intestinal in vitro digestion” procedure (Section 2.4). The readers are directed to the appropriate reference for the actual procedure in the revised manuscript (see Lines 152-154).
We have also reduced the information to describe the “Total starch measurement of undigested legumes using glucose oxidase/peroxidase (GOPOD) assay” (Section 2.5). The readers are directed to the appropriate reference [1] for the actual procedure in the revised manuscript (see Lines 173-175, 181-182).
We have also removed unnecessary introductory information for Section 2.7.
Point 4. Table 2: What is a potential reason that digested starch was not determined in chickpea at 30’?
Response: The processing of legumes has been conducted simultaneously as a big production line where samples were taken as a function of predefined processing time (up to 2 h) and then proceed to 6 h long in vitro digestion on the same day. Unfortunately, an accident took place during the trial resulting in the loss of chickpea sample processed for 30 min. Unfortunately, we could not produce chickpea processed for 30 min without repeating the entire series due to limited amount of chickpeas that we had. We acknowledge that this is not ideal. Based on the data, we found no or little change in digestion of chickpeas after being cooked less than 1 hour (see Table 2, no significant difference in tc for chickpea cooked for 15 min compared to 45 min.). We are confident that the trend and insight obtained in this work would not be compromised with the missing data points at 30 min of cooking.
Point 5. L667-674: Example of the introductory paragraph that can be omitted (details are in the material and methods section).
Response: Introductory paragraphs are removed. For other similar, introductory paragraphs/texts are removed from the revised manuscript.
Point 6. Figure 3 – Figure Error?
Response: Thank you for pointing out the mistake, we have now included the correct figure number (see Line 474).
Point 7. Figures 3 and 4 can be combined.
Response: We have combined both figures as Figure 3 with planes (a) and (b). We have also revised caption for Figure 3 (See Lines 470-474) and the correct figure number is used in the text when referring to these two figures.
Point 8. Figures 5 and 6 can be combined with planes a, b, c, d.
Response: The figures have been combined with the planes labelled appropriately. It is now referred as Figure 4. We have also revised caption for Figure 4 (See Lines 792-796) and the correct figure number is used in the text when referring to these four figures.
Point 9. L880-896 – Limitation and future directions – this section mainly summarizes the data and can be delated. Some future perspectives can be mentioned in conclusion.
Response: The section has been removed, with information on future perspectives added into the conclusion (see Lines 807-810).
Reviewer 2 Report
It is an interesting manuscript deserving, subject to minor revision.
The methods used to evaluate used to evaluate Starch, protein, lipid and moisture content of each legume should be mentioned and all the equipment need to have the following details:
Model #,
Equipment manufacture's name,
City of the manufacturer,
Country of manufacturer.
The producers of the different legumes should be reported.
Author Response
Thank you for your comment. The information regarding the equipment to evaluate starch, protein, lipid and moisture content has been added (see Lines 89-93).
Unfortunately, we were unable to track back the details of the producers of each different legume used in this study. Commercially available legumes within the New Zealand market come from the international trading distribution channel and to limit the variation, one big batch of the legumes was used in this study and purchased through a reliable retail channel. We acknowledge that this is the limitation of the current study and it opens an opportunity for future investigations to directly source legumes from producers.